# Optimal Robust PID Control for First- and Second-Order Plus Dead-Time Processes

**Takao Sato [1,*] , Itaru Hayashi [1], Yohei Horibe [1], Ramon Vilanova [2] and Yasuo Konishi [1]**

[1]  Department of Mechanical Engineering, University of Hyogo, Hyogo 651-2103, Japan;
   arieshorn9474@gmail.com (I.H.); table-y-tennis@i.softbank.jp (Y.H.); konishi@eng.u-hyogo.ac.jp (Y.K.)

[2]  Department of Telecommunications and Systems Engineering, Universitat Autònoma de Barcelona,
   08193 Barcelona, Spain; ramon.vilanova@uab.cat

*  Correspondence: tsato@eng.u-hyogo.ac.jp; Tel.: +81-792674983

**Abstract:** The present study proposes a new design method for a proportional-integral-derivative (PID) control system for first-order plus dead-time (FOPDT) and over-damped second-order plus dead-time (SOPDT) systems. What is presented is an optimal PID tuning constrained to robust stability. The optimal tuning is defined for each one of the two operation modes the control system may operate in: servo (reference tracking) and regulation (disturbance rejection). The optimization problem is stated for a normalized second-order plant that unifies FOPDT and SOPDT process models. Different robustness levels are considered and for each one of them, the set of optimal controller parameters is obtained. In a second step, suitable formulas are found that provide continuous values for the controller parameters. Finally, the effectiveness of the proposed method is confirmed through numerical examples.

**Keywords:** PID control; FOPDT system; SOPDT system; over-damping; sensitivity function; robust stability

## 1. Introduction

Proportional-integral-derivative (PID) control has been widely used because the control structure is simple and the role of tuning parameters is clear. See [1] as example of complex industrial application. The epoch-making rule of PID control has been investigated as the Ziegler and Nichols (ZN) method [2] and has been studied for more than half a century [3–6].

In the control system, stability is crucially important, and therefore, robust PID control has been studied [7,8]. Söylemez et al. have proposed a fast calculation method [9]. In this method, the stabilizing regions in the PID parameter space are decided for a fixed proportional gain, and the boundaries for stabilizing the proportional gain are also provided. Works [10–12] have proposed methods for the computation of all stabilizing PI(D) controllers, where the stability regions of PI(D) controllers are given. In [10], stabilizing PI controllers that achieve user-specified gain and phase margins are studied, and [11] gives the computation of stability boundaries of PI controller parameters that guarantee stability. Matušů and Prokop have proposed a method for the computation of all possible robustly stabilizing PID controllers for interval plants [10]. On the other hand, tracking performance also has to be improved. Since stability and tracking performance are in a trade-off relationship, a trade-off design method is required [13–15]. However, these trade-off design methods require complex calculations and so are unsuitable for industrial engineers.

A robust PID design method has been proposed [16,17] in which the tracking performance is optimized subject to the prescribed stability margin. Because of its usefulness, the unified design method has been extended from the first-order plus dead-time (FOPDT) system to the second-order

plus dead-time (SOPDT) system [18], and a two-degrees-of-freedom (2DoF) design has also been proposed [19]. Since these unified design methods are designed based on normalized plant models, the unified approach does not need a different solution whenever a different control system is designed.

The present method is inspired by the work presented in [18]. In this work, both FOPDT and SOPDT are considered. In a unified way, both process models' structures are considered and an optimal PID controller is adjusted for servo tracking and the disturbance rejection. However, in this approach, robustness is considered by simply adjusting the controller proportional gain. In the current work, a similar problem is faced, but the robustness is introduced in its full affectation of the controller parameters. Obviously, this affects the expression of all three controller parameters but allows for the improvement of performance. The rest of the paper is structured as follows. In Section 2, the controlled system model and the design objective are described. Section 3 gives the proposed PID tuning method, and the effectiveness of the proposed interpolation method is confirmed through numerical examples in Section 4. Finally, concluding remarks are presented in Section 5.

## 2. Control System and Design Objective

In this section, the optimization problem and control problem elements are presented. First of all, the process model and controller equations are considered, followed by the optimization function and robustness constraint. Prior to performing the optimization and presenting the corresponding results, the process model and controller equations are presented in normalized form.

### 2.1. Plant and Controller

A controlled plant is expressed as an over-damped second-order plus dead-time system given by the following transfer function:

$$P(s) = \frac{K}{(Ts+1)(aTs+1)} e^{-Ls}, \tag{1}$$

where $K$, $L$, and $T$ are the gain, the dead-time, and the time constant, respectively. Moreover, $a$ denotes the ratio of two time constants, and its range is $0 \leq a \leq 1$. It is worth noting that with the specific choice $a = 0$, the FOPDT case is covered.

This plant is controlled by a one-degree-of-freedom (1DoF) PID control law defined as follows:

$$U(s) = C_1(s)E(s) - C_2(s)Y(s) \tag{2}$$

$$C_1(s) = K_p \left( 1 + \frac{1}{T_i s} \right), \tag{3}$$

$$C_2(s) = K_p \left( \frac{T_d s}{(T_d/N)s + 1} \right), \tag{4}$$

$$E(s) = R(s) - Y(s),$$

where $U(s)$, $Y(s)$, and $R(s)$ are the control input, the plant output, and the reference input, respectively. In the PID control law, $K_p$, $T_i$, and $T_d$ are the proportional gain, the integral time, and the derivative time, respectively. The derivative compensation just affects the feedback signal and is filtered by a first-order transfer function to prevent the noise effect, in which the range of filter constant $N$ is usually from 5 to 33 [5]. Here, $N$ is set to 10 without loss of generality. The design objective of the present study is to decide the PID parameters such that a performance function is minimized, subject to the prescribed robust stability.

### 2.2. Performance Function and Constraint

First, the integral of absolute error (IAE) is used in the performance function, which is defined as follows:

$$J = \int_0^\infty |e(t)|dt = \int_0^\infty |r(t) - y(t)|dt. \tag{5}$$

Next, the constraint condition for assigning robust stability is also introduced. The sensitivity function of the discussed system, $S(s)$, is defined as follows:

$$S(s) = \frac{1}{1 + P(s)C(s)}, \tag{6}$$

$$C(s) = C_1(s) + C_2(s). \tag{7}$$

The maximum value of Equation (6), $M_s$, is decided as follows:

$$M_s = \max_\omega \frac{1}{|1 + P(j\omega)C(j\omega)|}. \tag{8}$$

The relationship between $M_s$ and the gain margin ($g_m$) and that between $M_s$ and the phase margin ($\phi_m$) are as follows [3,4]:

$$g_m \geq \frac{M_s}{M_s - 1}, \tag{9}$$

$$\phi_m \geq 2\arcsin\frac{1}{2M_s}. \tag{10}$$

Therefore, robust stability is achieved by satisfying the following constraint:

$$|M_s^d - M_s| \cong 0, \tag{11}$$

where $M_s^d$ denotes the desired (prescribed) robustness. In the present study, the prescribed robustness is selected in the rage from 1.4 to 2.0 [18,19].

### 2.3. Normalized Process Model and Controller

In the proposed method, the controller parameters are optimized for the following normalized controlled plant:

$$\hat{P}(\hat{s}) = \frac{1}{(\hat{s}+1)(a\hat{s}+1)}e^{-\tau_0\hat{s}}, \tag{12}$$

$$\hat{C}_1(\hat{s}) = \kappa_p\left(1 + \frac{1}{\tau_i\hat{s}}\right), \tag{13}$$

$$\hat{C}_2(\hat{s}) = \kappa_p\left(\frac{\tau_d\hat{s}}{(\tau_d/N)\hat{s}+1}\right). \tag{14}$$

Here, $\kappa_p$, $\tau_i$, and $\tau_d$ are the following normalized PID parameters:

$$\kappa_p = K_pK, \tag{15}$$

$$\tau_i = \frac{T_i}{T}, \tag{16}$$

$$\tau_d = \frac{T_d}{T}, \tag{17}$$

and $\tau_0(= L/T)$ is the normalized dead time.

In the proposed method, in order to optimize the performance with respect to tracking and disturbance error such that the prescribed robust stability is achieved, the normalized PID parameters are decided based on the constrained optimization problem so that Equation (5) is optimized subject to Equation (11).

## 3. Unified Optimal PID Tuning with Robust Stability

In the present study, the normalized PID parameters are optimized for the servo and regulation performances, where $a$ is set to be from 0 to 1.0 in increments of 0.1, and $\tau_0$ is set to be from 0.2 to 2.0 in increments of 0.1.

In the present study, the optimal parameters are obtained numerically with the MATLAB function *fmincon* (Mathworks, Inc., Natuck, MA, USA), where the *interior-point* algorithm is used and the constraint $M_s^d$ is set to 1.4, 1.6, 1.8, and 2.0. In the servo optimization process, the reference input $r(t)$ is set as a unit step function, and the disturbance is not given. On the other hand, in the regulation optimization process, as a unit step function, the distrubance $d(t)$ is added to the control input, and the reference input is set to 0.

### 3.1. Servo Optimization

The optimized normalized PID parameters for the servo tracking optimization in the given finite plants are plotted as ○ symbols in Figures 1–3.

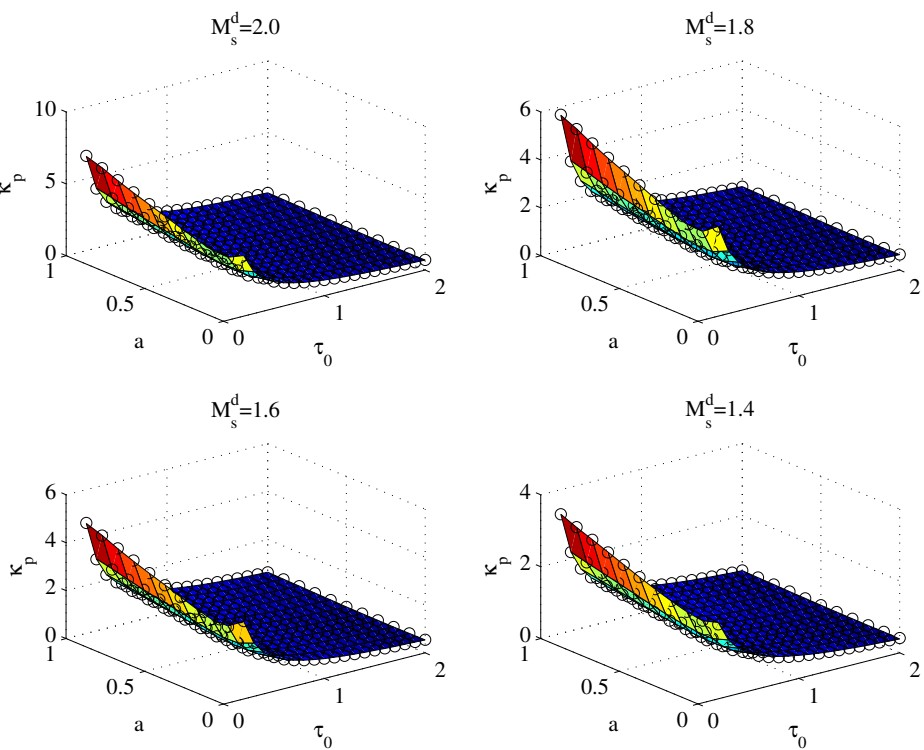

**Figure 1.** Calculated and approximated optimal $\kappa_p$ in the servo mode.

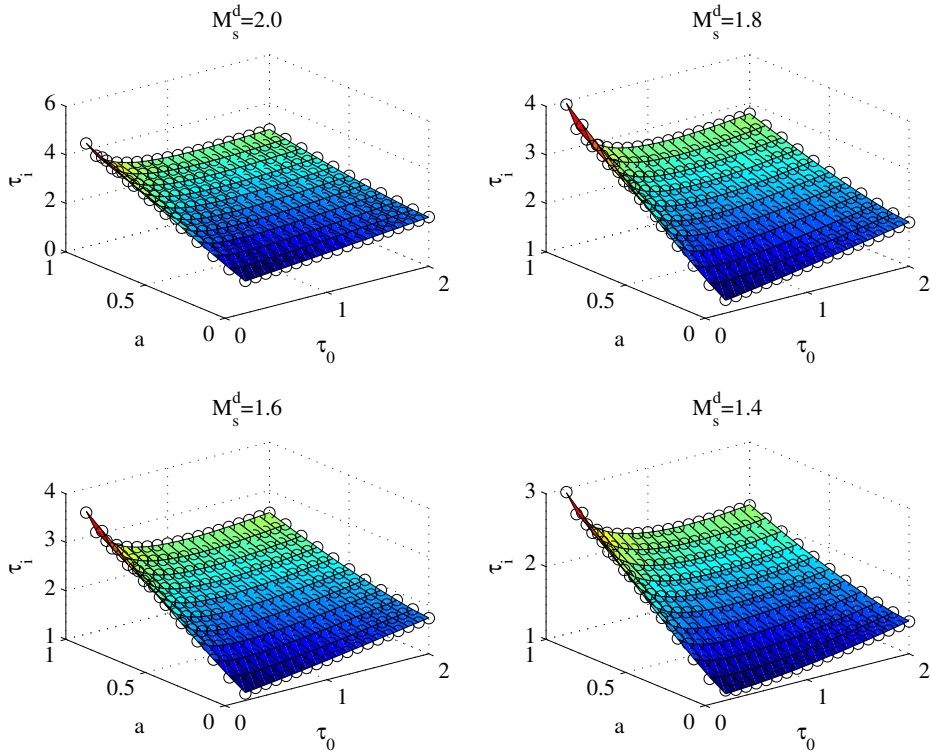

**Figure 2.** Calculated and approximated optimal $\tau_i$ in the servo mode.

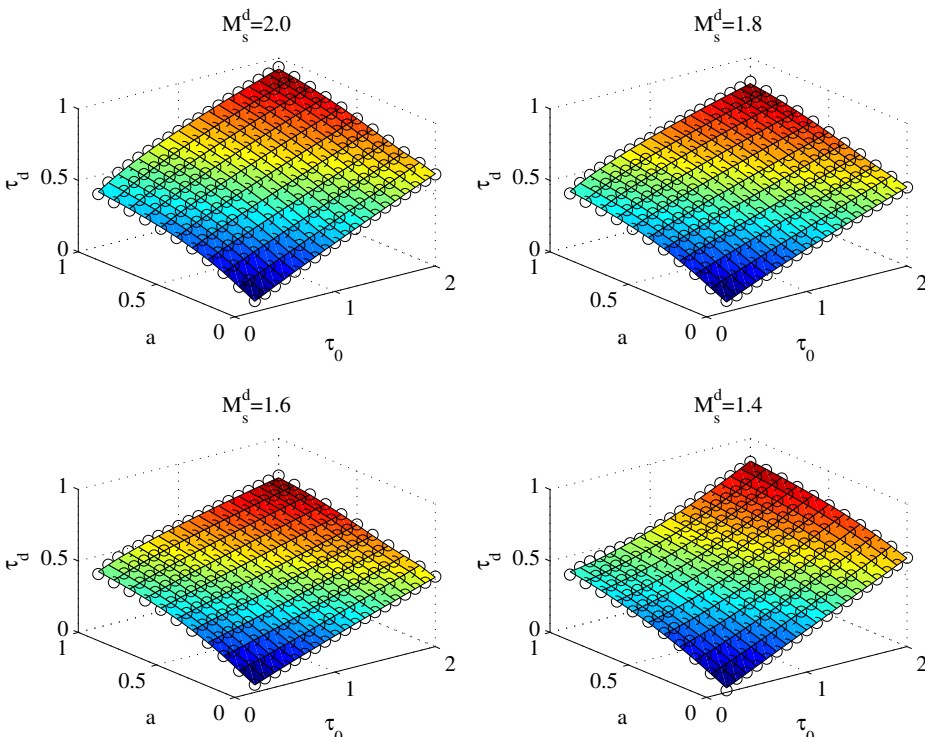

**Figure 3.** Calculated and approximated optimal $\tau_d$ in the servo mode.

To obtain the unified design method, the obtained normalized PID parameters are approximated by the following equations:

$$\kappa_p(a, \tau_0, M_s^d) = a_0(a, M_s^d) + a_1(a, M_s^d)\tau_0^{a_2(a, M_s^d)}, \tag{18}$$

$$a_0(a, M_s^d) = \frac{\alpha_0(M_s^d) + \alpha_1(M_s^d)a + \alpha_2(M_s^d)a^2}{\alpha_3(M_s^d) + a},$$

$$a_1(a, M_s^d) = \frac{\alpha_4(M_s^d) + \alpha_5(M_s^d)a + \alpha_6(M_s^d)a^2}{\alpha_7(M_s^d) + a},$$

$$a_3(a, M_s^d) = \alpha_8(M_s^d)a^5 + \alpha_9(M_s^d)a^4 + \alpha_{10}(M_s^d)a^3,$$
$$+ \quad \alpha_{11}(M_s^d)a^2 + \alpha_{12}a + \alpha_{13},$$

$$\tau_i(a, \tau_0, M_s^d) = b_0(\tau_0, M_s^d)a^{b_1(\tau_0, M_s^d)} + b_2(\tau_0, M_s^d), \tag{19}$$

$$b_0(\tau_0, M_s^d) = \beta_0(M_s^d) + \beta_1(M_s^d)\tau_0^{\beta_2(M_s^d)},$$

$$b_1(\tau_0, M_s^d) = \beta_3(M_s^d) + \beta_4(M_s^d)\tau_0^{\beta_5(M_s^d)},$$

$$b_2(\tau_0, M_s^d) = \beta_6(M_s^d) + \beta_7(M_s^d)\tau_0^{\beta_8(M_s^d)},$$

$$\tau_d(a, \tau_0, M_s^d) = c_0(a, M_s^d) + c_1(a, M_s^d)\tau_0^{C_2(a, M_s^d)}, \tag{20}$$

$$c_0(a, M_s^d) = \frac{\gamma_0(M_s^d) + \gamma_1(M_s^d)a + \gamma_2(M_s^d)a^2}{\gamma_3(M_s^d) + a},$$

$$c_1(a, M_s^d) = \frac{\gamma_4(M_s^d) + \gamma_5(M_s^d)a + \gamma_6(M_s^d)a^2}{\gamma_7(M_s^d) + a},$$

$$c_2(a, M_s^d) = \frac{\gamma_8(M_s^d) + \gamma_9(M_s^d)a + \gamma_{10}(M_s^d)a^2}{\gamma_{11}(M_s^d) + a},$$

where the coefficient parameters are shown in Tables 1–3.

**Table 1.** Coefficient parameters $\alpha_i(M_s^d)$ for $\kappa_p$ in the servo mode.

| $M_s^d$ | 1.4 | 1.6 | 1.8 | 2.0 |
|---|---|---|---|---|
| $\alpha_0$ | 0.001519 | 1.484 | 0.1135 | 0.2830 |
| $\alpha_1$ | 0.2972 | 1.112 | 0.5295 | 0.72443 |
| $\alpha_2$ | $-0.06124$ | $-0.3106$ | $-0.04377$ | $-0.1049$ |
| $\alpha_3$ | 0.05894 | 5.716 | 0.4141 | 0.9061 |
| $\alpha_4$ | 0.06374 | 0.2580 | 0.2712 | 0.3628 |
| $\alpha_5$ | 0.1485 | 0.2368 | 0.1780 | 0.1615 |
| $\alpha_6$ | 0.3625 | 0.4666 | 0.5937 | 0.6923 |
| $\alpha_7$ | 0.1036 | 0.4520 | 0.3751 | 0.4683 |
| $\alpha_8$ | $-1.897$ | 4.867 | 4.407 | 5.195 |
| $\alpha_9$ | 6.302 | $-14.28$ | $-12.85$ | $-15.25$ |
| $\alpha_{10}$ | $-8.412$ | 15.44 | 13.64 | 16.41 |
| $\alpha_{11}$ | 5.821 | $-7.242$ | $-5.972$ | $-7.434$ |
| $\alpha_{12}$ | $-2.157$ | 1.079 | 0.5559 | 0.8462 |
| $\alpha_{13}$ | $-0.7626$ | $-1.026$ | $-0.9744$ | $-1.004$ |

**Table 2.** Coefficient parameters $\beta_i(M_s^d)$ for $\tau_i$ in the servo mode.

| $M_s^d$ | 1.4 | 1.6 | 1.8 | 2.0 |
|---|---|---|---|---|
| $\beta_0$ | 0.1221 | 0.4341 | 0.4868 | 0.3670 |
| $\beta_1$ | 0.7921 | 0.6098 | 0.6553 | 0.8488 |
| $\beta_2$ | $-0.4862$ | $-0.7187$ | $-0.7553$ | $-0.6927$ |
| $\beta_3$ | $-0.6852$ | $-0.7298$ | $-0.7304$ | $-5.795$ |
| $\beta_4$ | 1.999 | 2.000 | 1.999 | 7.101 |
| $\beta_5$ | 0.06147 | 0.05941 | 0.05342 | 0.02174 |
| $\beta_6$ | 1.082 | 1.118 | 1.208 | 1.245 |
| $\beta_7$ | 0.1174 | 0.2808 | 0.3231 | 0.3738 |
| $\beta_8$ | 1.659 | 1.149 | 1.125 | 1.061 |

**Table 3.** Coefficient parameters $\gamma_i(M_s^d)$ for $\tau_d$ in the servo mode.

| $M_s^d$ | 1.4 | 1.6 | 1.8 | 2.0 |
|---|---|---|---|---|
| $\gamma_0$ | $-0.07558$ | $-0.01783$ | $-0.01342$ | $-0.006020$ |
| $\gamma_1$ | 1.461 | 1.672 | 1.542 | 0.8320 |
| $\gamma_2$ | $-0.2621$ | $-0.7720$ | $-0.7770$ | $-0.3588$ |
| $\gamma_3$ | 1.986 | 1.993 | 1.959 | 0.9935 |
| $\gamma_4$ | 0.2557 | 0.1633 | 0.1825 | 0.1117 |
| $\gamma_5$ | 0.002213 | $-0.1014$ | 0.003464 | 0.1548 |
| $\gamma_6$ | 0.02149 | 0.3830 | 0.4003 | 0.3101 |
| $\gamma_7$ | 0.7684 | 0.6222 | 0.5993 | 0.3231 |
| $\gamma_8$ | 0.8714 | 0.2037 | 0.1547 | 0.04407 |
| $\gamma_9$ | 1.999 | 1.497 | 1.184 | 1.072 |
| $\gamma_{10}$ | 0.08513 | $-0.8993$ | $-0.5988$ | $-0.4221$ |
| $\gamma_{11}$ | 0.8892 | 0.2228 | 0.1772 | 0.04822 |

The uncalculated normalized PID parameters can be interpolated using Equations (18)–(20) and are plotted as colored surfaces in Figures 1–3.

The values of $M_s$ for $M_s^d = 1.4$, 1.6, 1.8, and 2.0 are plotted in Figure 4, and the obtained maximum, average, and minimum values for $M_s$ are listed in Table 4. As shown in this table, the errors for the prescribed maximum sensitivity are within $\pm 3\%$.

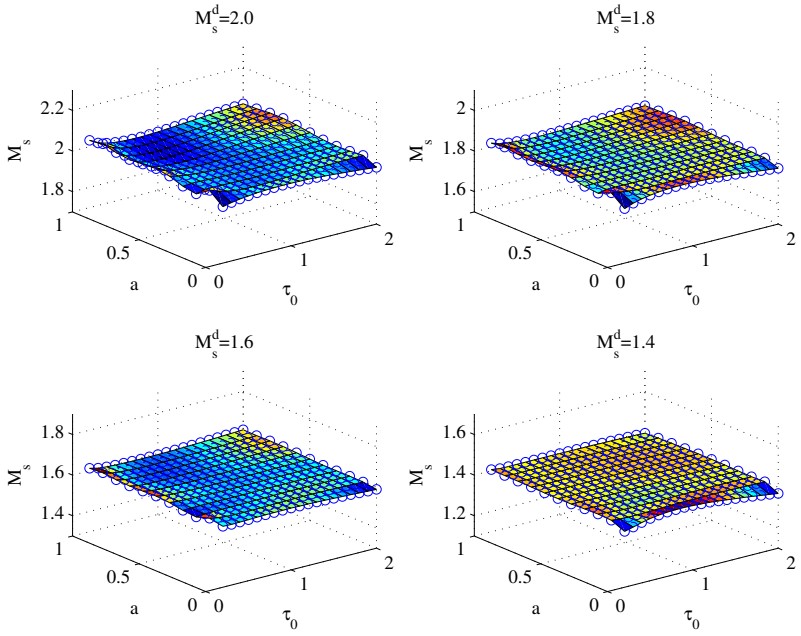

**Figure 4.** $M_s$ in the servo mode.

**Table 4.** Evaluation of $|M_s - M_s^d|$ in the servo mode.

| $M_s$ | 1.4 | 1.6 | 1.8 | 2.0 |
|---|---|---|---|---|
| max | 1.418 | 1.618 | 1.826 | 2.042 |
| ave | 1.400 | 1.600 | 1.800 | 2.001 |
| min | 1.369 | 1.589 | 1.768 | 1.978 |

*3.2. Regulation Optimization*

In a similar way to the servo optimization, the optimized normalized PID parameters for disturbance rejection optimization are shown in Figures 5–7, where uncalculated normalized PID parameters are interpolated by Equations (21)–(23).

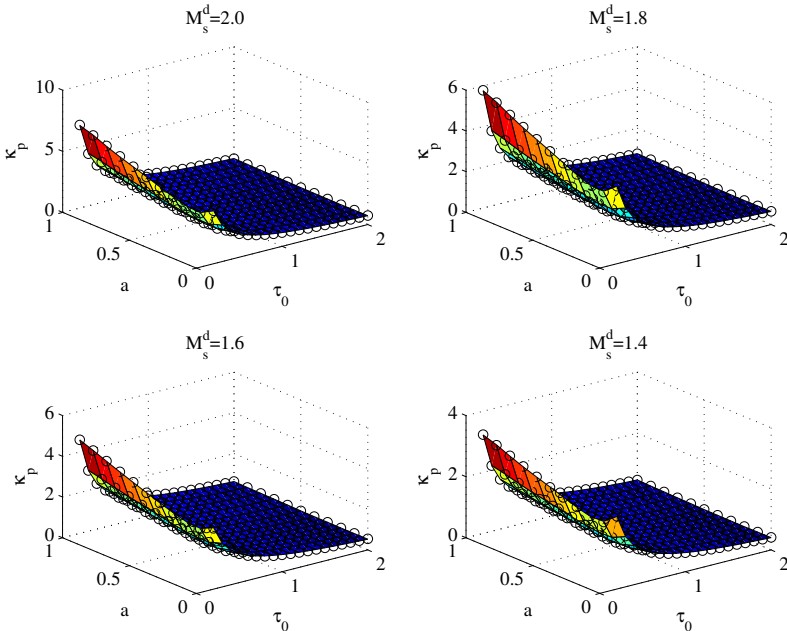

**Figure 5.** Calculated and approximated optimal $\kappa_p$ in the regulation mode.

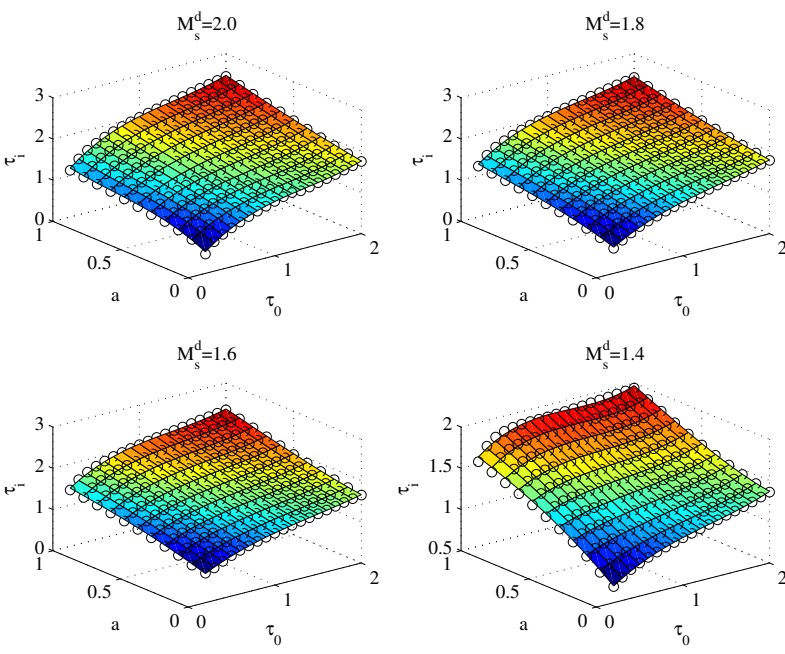

**Figure 6.** Calculated and approximated optimal $\tau_i$ in the regulation mode.

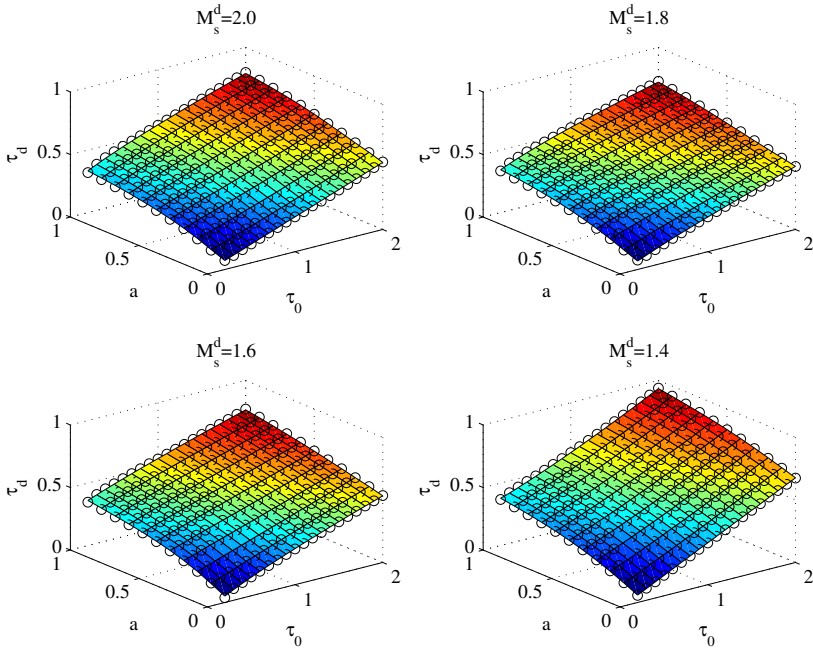

**Figure 7.** Calculated and approximated optimal $\tau_d$ in the regulation mode.

$$\kappa_p(a, \tau_0, M_s^d) \quad = \quad a_0(a, M_s^d) + a_1(a, M_s^d)\tau_0^{a_2(a, M_s^d)}, \tag{21}$$

$$a_0(a, M_s^d) \quad = \quad \frac{\alpha_0(M_s^d) + \alpha_1(M_s^d)a + \alpha_2(M_s^d)a^2}{\alpha_3(M_s^d) + a},$$

$$a_1(a, M_s^d) \quad = \quad \frac{\alpha_4(M_s^d) + \alpha_5(M_s^d)a + \alpha_6(M_s^d)a^2}{\alpha_7(M_s^d) + a},$$

$$a_3(a, M_s^d) \quad = \quad \alpha_8(M_s^d)a^5 + \alpha_9(M_s^d)a^4 + \alpha_{10}(M_s^d)a^3$$

$$+ \quad \alpha_{11}(M_s^d)a^2 + \alpha_{12}a + \alpha_{13},$$

$$\tau_i(a, \tau_0, M_s^d) \quad = \quad b_0(\tau_0, M_s^d) + b_1(\tau_0, M_s^d)\tau_0,$$

$$+ \quad b_2(\tau_0, M_s^d)\tau_0^2 + b_3(\tau_0, M_s^d)\tau_0^3 , \tag{22}$$

$$b_0(\tau_0, M_s^d) \quad = \quad \beta_0(M_s^d) + \beta_1(M_s^d)\tau_0,$$

$$+ \quad \beta_2(M_s^d)\tau_0^2 + \beta_3(M_s^d)\tau_0^3 ,$$

$$b_1(\tau_0, M_s^d) \quad = \quad \beta_4(M_s^d) + \beta_5(M_s^d)\tau_0,$$

$$+ \quad \beta_6(M_s^d)\tau_0^2 + \beta_7(M_s^d)\tau_0^3 ,$$

$$b_2(\tau_0, M_s^d) \quad = \quad \beta_8(M_s^d) + \beta_9(M_s^d)\tau_0,$$

$$+ \quad \beta_{10}(M_s^d)\tau_0^2 + \beta_{11}(M_s^d)\tau_0^3 ,$$

$$b_3(\tau_0, M_s^d) \quad = \quad \beta_{12}(M_s^d) + \beta_{13}(M_s^d)\tau_0,$$

$$+ \quad \beta_{14}(M_s^d)\tau_0^2 + \beta_{15}(M_s^d)\tau_0^3 ,$$

$$\tau_d(a, \tau_0, M_s^d) = c_0(a, M_s^d) + c_1(a, M_s^d)\tau_0^{C_2(a, M_s^d)},$$

$$c_0(a, M_s^d) = \frac{\gamma_0(M_s^d) + \gamma_1(M_s^d)a + \gamma_2(M_s^d)a^2}{\gamma_3(M_s^d) + a},$$

$$c_1(a, M_s^d) = \frac{\gamma_4(M_s^d) + \gamma_5(M_s^d)a + \gamma_6(M_s^d)a^2}{\gamma_7(M_s^d) + a},$$

$$c_2(a, M_s^d) = \frac{\gamma_8(M_s^d) + \gamma_9(M_s^d)a + \gamma_{10}(M_s^d)a^2}{\gamma_{11}(M_s^d) + a},$$

(23)

where the coefficient parameters are shown in Tables 5–7.

**Table 5.** Coefficient parameters $\alpha_i(M_s^d)$ for $\kappa_p$ in the regulation mode.

| $M_s^d$ | 1.4 | 1.6 | 1.8 | 2.0 |
|---|---|---|---|---|
| $\alpha_0$ | 0.1275 | 0.01763 | 0.2852 | 0.4909 |
| $\alpha_1$ | 0.3274 | 0.3660 | 0.6534 | 0.8894 |
| $\alpha_2$ | $-0.06243$ | $-0.007755$ | $-0.1128$ | $-0.1818$ |
| $\alpha_3$ | 0.7013 | 0.1065 | 0.9467 | 1.422 |
| $\alpha_4$ | 0.1858 | 0.1245 | 0.3214 | 0.3872 |
| $\alpha_5$ | 0.1481 | 0.2176 | 0.1657 | 0.1664 |
| $\alpha_6$ | 0.3932 | 0.4302 | 0.6290 | 0.7286 |
| $\alpha_7$ | 0.4191 | 0.1845 | 0.4653 | 0.5007 |
| $\alpha_8$ | 4.520 | 1.649 | 4.981 | 5.511 |
| $\alpha_9$ | $-13.19$ | $-4.542$ | $-14.60$ | $-16.06$ |
| $\alpha_{10}$ | 14.12 | 4.225 | 15.67 | 17.15 |
| $\alpha_{11}$ | $-6.441$ | $-1.104$ | $-7.098$ | $-7.721$ |
| $\alpha_{12}$ | 0.8919 | $-0.5000$ | 0.8399 | 0.9067 |
| $\alpha_{13}$ | $-0.9939$ | $-0.8934$ | $-1.008$ | $-1.017$ |

**Table 6.** Coefficient parameters $\beta_i(M_s^d)$ for $\tau_i$ in the regulation mode.

| $M_s^d$ | 1.4 | 1.6 | 1.8 | 2.0 |
|---|---|---|---|---|
| $\beta_0$ | 0.5352 | 0.4780 | 0.3580 | 0.1296 |
| $\beta_1$ | 1.842 | 1.202 | 1.343 | 2.258 |
| $\beta_2$ | $-1.538$ | $-0.8566$ | $-1.399$ | $-2.914$ |
| $\beta_3$ | 0.5307 | 0.2571 | 0.6334 | 1.320 |
| $\beta_4$ | 0.9230 | 1.248 | 1.522 | 2.219 |
| $\beta_5$ | $-2.976$ | $-0.8036$ | $-0.9822$ | $-3.100$ |
| $\beta_6$ | 5.838 | 2.361 | 2.878 | 4.805 |
| $\beta_7$ | $-2.529$ | $-0.9300$ | $-1.452$ | $-1.701$ |
| $\beta_8$ | $-0.5116$ | $-0.6027$ | $-0.7495$ | $-1.370$ |
| $\beta_9$ | 1.934 | 0.1011 | 0.3725 | 1.567 |
| $\beta_{10}$ | $-4.584$ | $-1.220$ | $-1.800$ | $-1.546$ |
| $\beta_{11}$ | 2.151 | 0.5330 | 1.039 | 0.02506 |
| $\beta_{12}$ | 0.1290 | 0.1366 | 0.1736 | 0.3349 |
| $\beta_{13}$ | $-0.4199$ | 0.05628 | $-0.05291$ | $-0.2219$ |
| $\beta_{14}$ | 1.090 | 0.1802 | 0.3860 | $-0.04209$ |
| $\beta_{15}$ | $-0.5327$ | $-0.08658$ | $-0.2399$ | 0.2453 |

**Table 7.** Coefficient parameters $\gamma_i(M_s^d)$ for $\tau_d$ in the regulation mode.

| $M_s^d$ | 1.4 | 1.6 | 1.8 | 2.0 |
|---|---|---|---|---|
| $\gamma_0$ | −0.01709 | −0.01810 | −0.007789 | 0.02155 |
| $\gamma_1$ | 0.9843 | 0.4629 | 1.402 | 0.6965 |
| $\gamma_2$ | −0.1369 | −0.004881 | −0.5348 | −0.1997 |
| $\gamma_3$ | 1.448 | 0.4799 | 2.351 | 1.073 |
| $\gamma_4$ | 0.1451 | 0.06998 | 0.1886 | 0.05419 |
| $\gamma_5$ | 0.2027 | 0.1237 | 0.06031 | 0.1531 |
| $\gamma_6$ | 0.02381 | 0.1178 | 0.2215 | 0.1676 |
| $\gamma_7$ | 0.4183 | 0.2145 | 0.6713 | 0.2098 |
| $\gamma_8$ | 0.3704 | 0.1160 | 0.1664 | 0.07437 |
| $\gamma_9$ | 1.271 | 1.191 | 1.026 | 1.235 |
| $\gamma_{10}$ | −0.01542 | −0.3177 | −0.3190 | −0.4418 |
| $\gamma_{11}$ | 0.3743 | 0.1438 | 0.1948 | 0.07446 |

The values of $M_s$ for $M_s^d = 1.4$, 1.6, 1.8, and 2.0 are plotted in Figure 8, and the obtained values of $M_s$ are listed in Table 8. Since the errors for the prescribed maximum sensitivity are within ±3%, the effectiveness of the proposed method is confirmed.

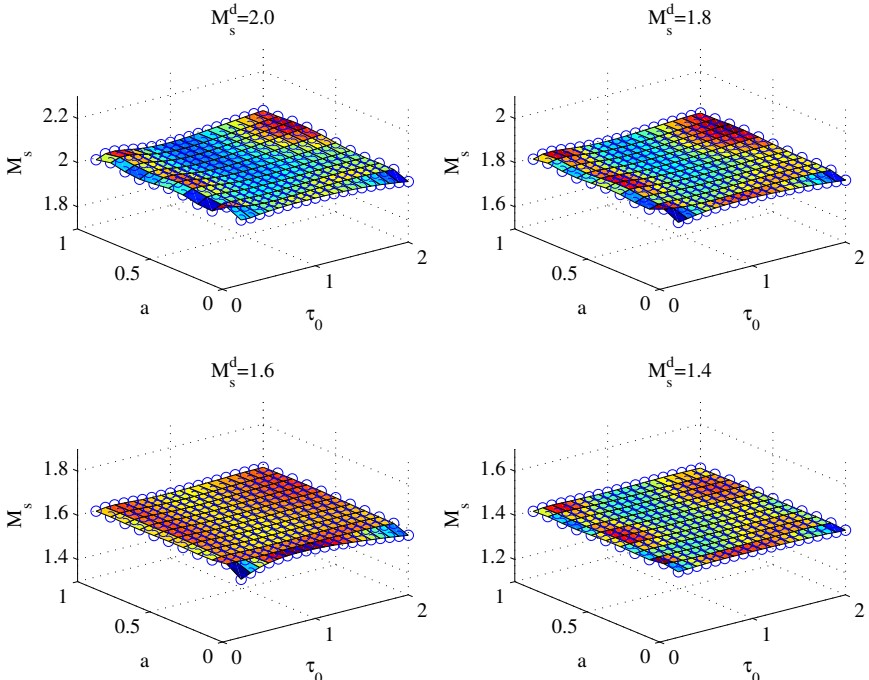

**Figure 8.** $M_s$ in the regulation mode.

**Table 8.** Evaluation of $|M_s - M_s^d|$ in the regulation mode.

| $M_s$ | 1.4 | 1.6 | 1.8 | 2.0 |
|---|---|---|---|---|
| max | 1.406 | 1.617 | 1.816 | 2.029 |
| ave | 1.400 | 1.600 | 1.800 | 2.001 |
| min | 1.392 | 1.562 | 1.780 | 1.974 |

From Tables 4–8, the prescribed robust stability is sufficiently achieved, even though the optimal trade-off PID parameters are easily implemented using the calculated look-up tables.

## 4. Numerical Simulation

### 4.1. Servo–Regulation Trade-off Performance for Various SOPDT Systems

The trade-off control performance between servo and regulation optimization is shown for each $M_s^d$, and the interpolation validity is confirmed for various values of $a$.

The controlled plant is given as follows:

$$P(s) = \frac{1}{(s+1)(as+1)}e^{-0.5s}, \tag{24}$$

where the proposed method is designed for $a = 0, 0.3, 0.5, 0.8,$ and $1.0$. Furthermore, the control system is optimized for the servo and regulation modes, and the control performances are compared. The reference input is set to 1.0, and the control input is disturbed by the step signal with amplitude 0.5 after 20 s. The simulated plant outputs are shown in Figures 9–13, where the solid and dashed lines are the servo and regulation modes, respectively, and the green, magenta, blue, and black lines are $M_s^d = 1.4, 1.6, 1.8,$ and $2.0$, respectively.

Figures 9–13 show that the tracking performance of the servo optimization tuning is superior to that of the regulation optimization tuning, and the regulating performances of the servo and regulation optimization tunings are reversed when the control input is disturbed. Furthermore, both the servo and regulation performances using large $M_s^d$ are superior to those using small $M_s^d$. Therefore, the trade-off tuning of the proposed method is confirmed.

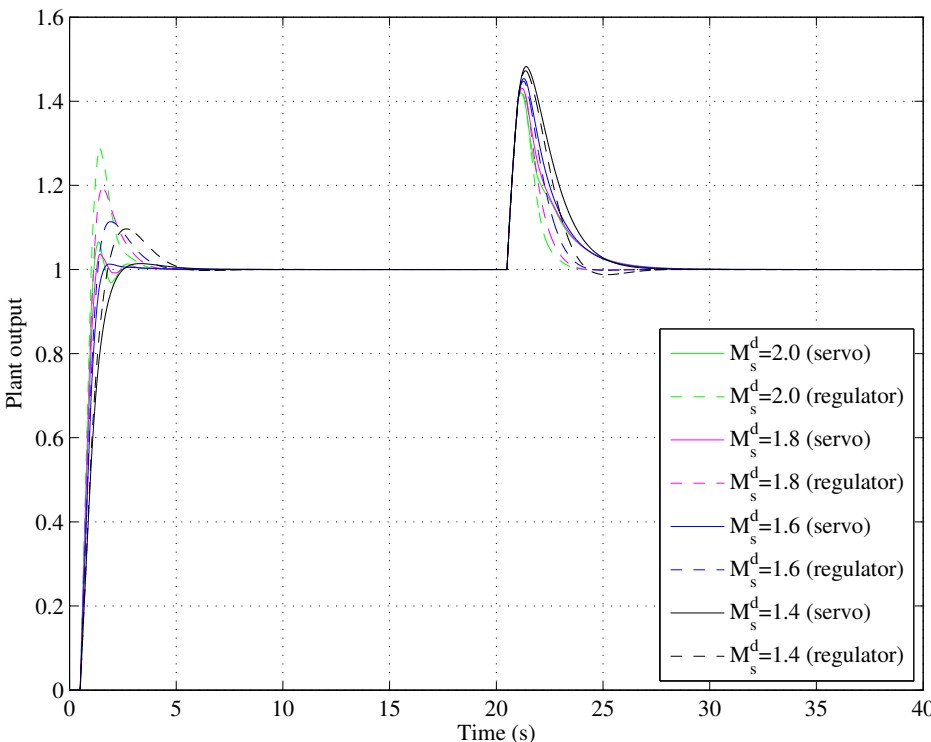

**Figure 9.** Servo and regulation performances for $a = 0$.

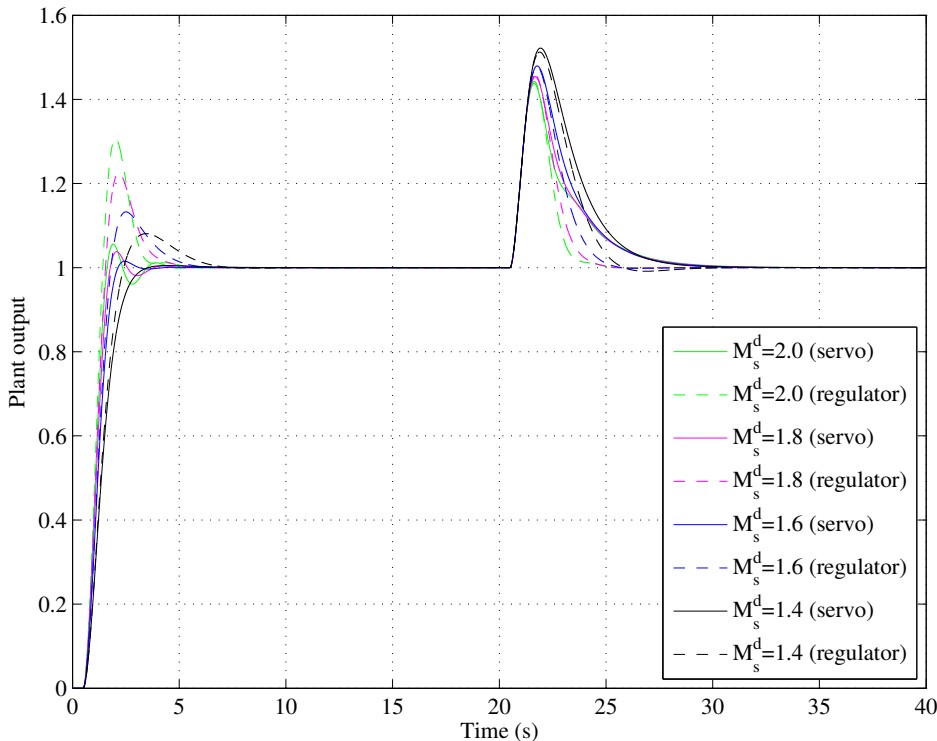

**Figure 10.** Servo and regulation performances for $a = 0.3$.

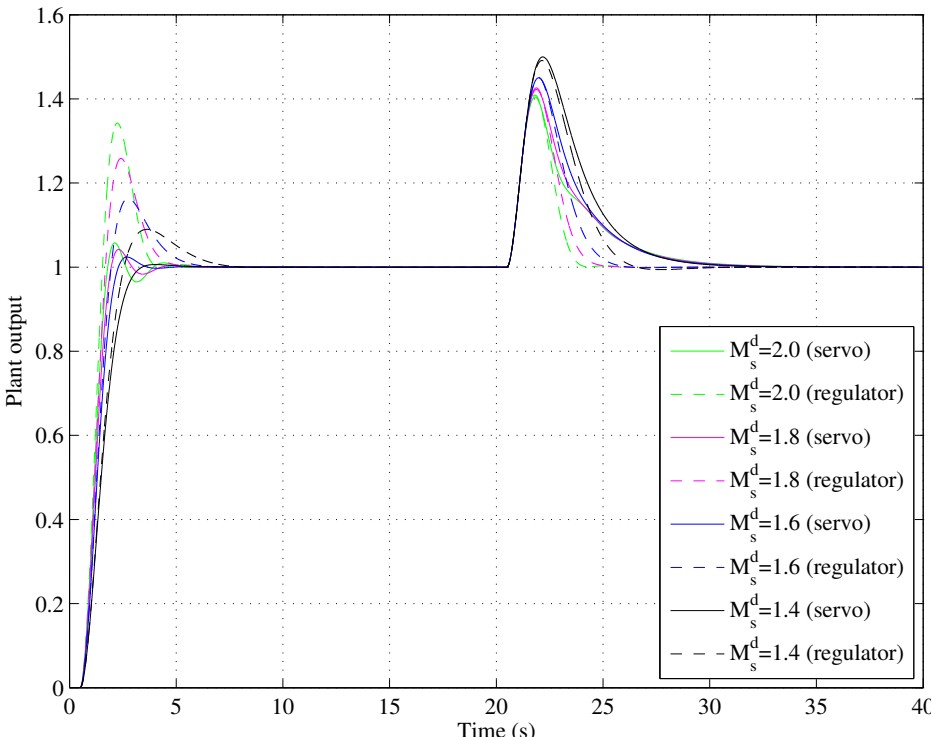

**Figure 11.** Servo and regulation performances for $a = 0.5$.

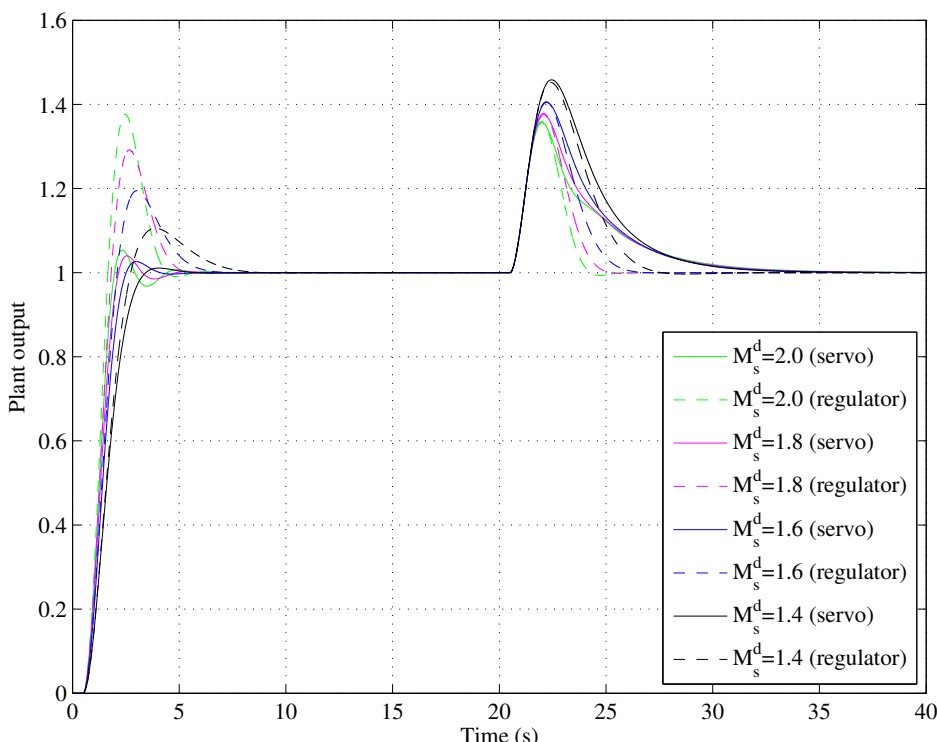

**Figure 12.** Servo and regulation performances for $a = 0.8$.

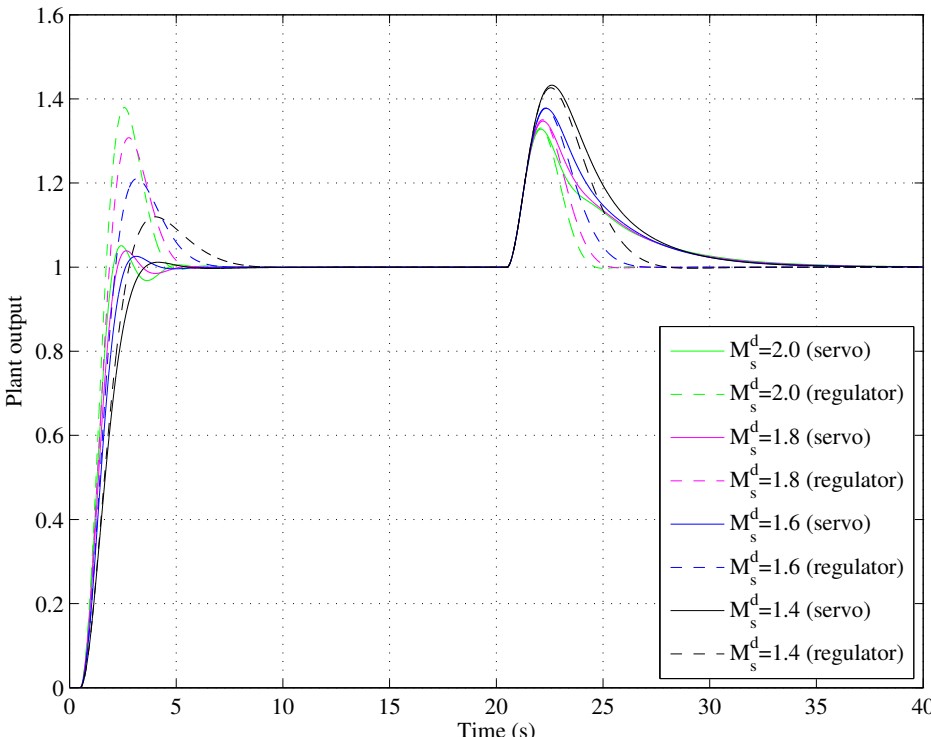

**Figure 13.** Servo and regulation performances for $a = 1.0$.

### 4.2. Tracking–Robustness Trade-Off Performance

The proposed servo- and regulation-optimized controllers are designed based on the following equation:

$$P(s) = \frac{1}{s+1}e^{-0.5s}. \tag{25}$$

Equation (25) is not the SOPDT system but the FOPDT system. This is because the proposed method is available for controlling the FOPDT system as well as the SOPDT. In order to confirm robust stability for the prescribed $M_s^d$, as plant perturbation occurs, the plant model is changed to Equation (26) after 60 s.

$$P'(s) = \frac{1.9}{(1.4s+1)(0.2 \times 1.4s+1)}e^{-0.7s}. \tag{26}$$

The control results using four $M_s^d$ tunings are shown in Figure 14, and the enlarged views for the tracking, regulation, and model perturbation trajectories are also shown in Figure 15. This figure shows that the smaller the $M_s^d$, the greater the robust stability because the plant output diverges when $M_s^d = 2.0$, and the plant output converges to the reference input using small $M_s^d$. Therefore, the achievement of the prescribed robust stability is confirmed using the proposed method.

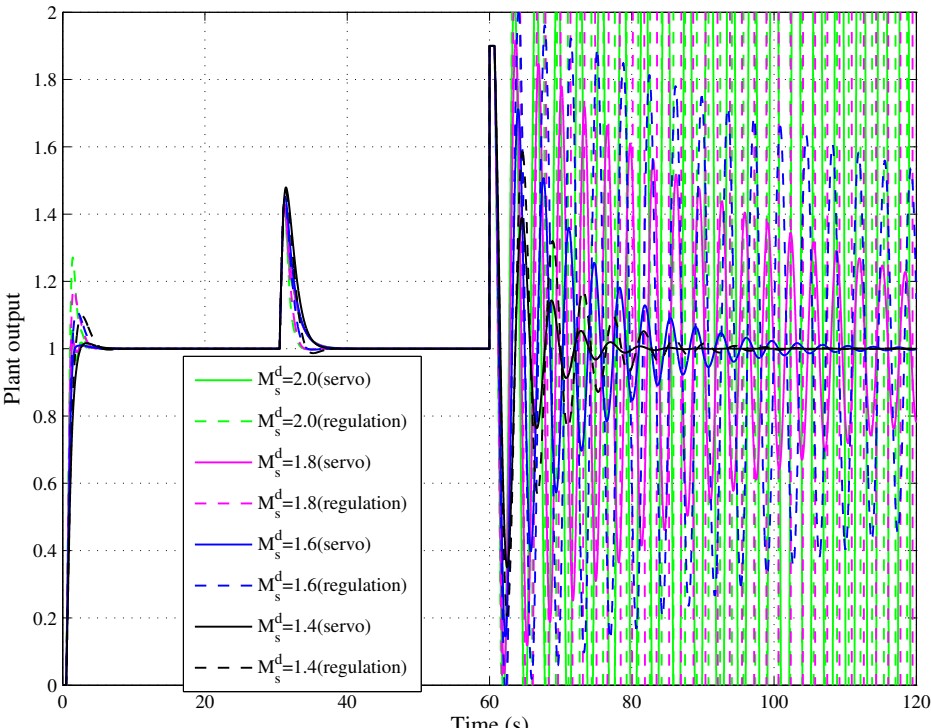

**Figure 14.** Model perturbation performance.

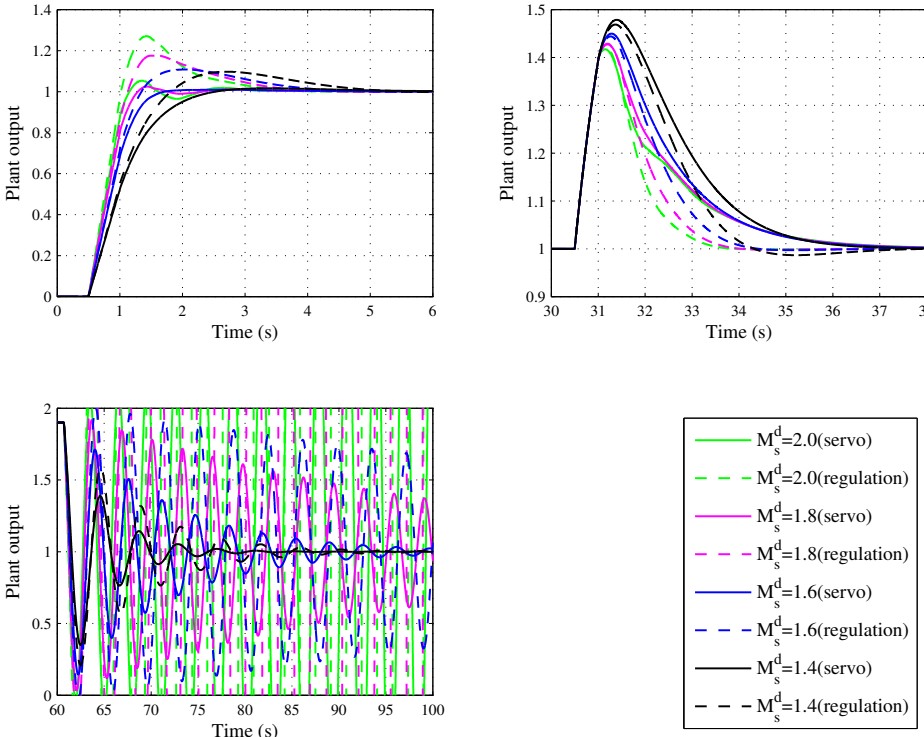

**Figure 15.** Enlarged views of Figure 14.

## 4.3. Comparison with Conventional Methods

The proposed method is compared with conventional methods: the ZN method [2], conventional SOPDT-based PI and PID control [18], and the conventional SOPDT-based 2DoF PI control [19].

As a controlled plant, consider the following transfer function:

$$P(s) = \frac{1}{(s+1)(0.62s+1)}e^{-1.5s}. \tag{27}$$

The reference input is 1.0, the control input is disturbed by the step signal with amplitude 0.5 after 150 s, and the controlled plant is perturbed to Equation (28) after 300 s.

$$P'(s) = \frac{1.4}{(1.2s+1)(0.36 \times 1.2s+1)}e^{-1.2s}. \tag{28}$$

In the design of the conventional SOPDT model-based methods [18,19], the robust stability is designed with $M_s^d = 1.6$, and $a = 0.5$ is used to decide the PID parameters because $a = 0.62$ in Equation (27) is not directly used. In the proposed method, $M_s^d$ is also set to 1.6 for comparison with the conventional methods. In the conventional [18] and the proposed trade-off design methods, the controller parameters are designed in the servo and regulation modes, respectively. Furthermore, only in the conventional 2DoF design method [19], 2DoF parameter $\beta$ is used in the following control law:

$$U(s) = K_p\left(\beta R(s) - Y(s) + \frac{1}{T_i s}(R(s) - Y(s)).\right) \tag{29}$$

The calculated controller parameters based on Equation (27) are shown in Table 9.

**Table 9.** Proportional-integral-derivative (PID) control system parameters.

| Design Method | $K_p$ | $T_i$ | $T_d$ | $\beta$ |
|---|---|---|---|---|
| ZN [2] | 1.52 | 3.44 | 0.859 | – |
| 1DoF PI (servo) [18] | 0.469 | 1.92 | – | – |
| 1DoF PI (reg) [18] | 0.488 | 2.09 | – | – |
| 1DoF PID (servo) [18] | 0.633 | 2.18 | 0.642 | – |
| 1DoF PID (reg) [18] | 0.610 | 1.68 | 0.774 | – |
| 2DoF PI [19] | 0.431 | 1.66 | – | 0.983 |
| Proposed 1DoF PID (servo) | 0.670 | 2.04 | 0.567 | – |
| Proposed 1DoF PID (reg) | 0.665 | 1.87 | 0.582 | – |

The control results are shown in Figure 16, and the enlarged views of the tracking, regulation, and model perturbation trajectories are also shown in Figure 17. Using the conventional [18,19] and proposed SOPDT-based design methods, the closed-loop system stabilizes even when the plant is perturbed after 300 s because the stability margin is designed. On the other hand, using the ZN method, the closed-loop system is unstable after plant perturbation, although it can be stabilized until plant perturbation.

The obtained control results are evaluated by the IAE, as shown in Table 10. In both the servo performance from to 0 s to 150 s and the regulation performance from 150 s to 300 s, the IAE of the ZN method is the largest among the considered methods because the plant output vibrates and because the settling time is the longest among the considered design methods.

Table 10 shows that the trade-off between the servo and regulation performances is achieved using the conventional [18] and proposed trade-off methods because in the servo evaluation period (0 s to 150 s), the IAE using the servo-optimized controllers, i.e., 1DoF PI (servo), 1DoF PID (servo), and the proposed 1DoF PID (servo), is smaller than that obtained using the regulation-optimized controllers, i.e., 1DoF PI (reg), 1DoF PID (reg), and the proposed 1DoF PID (reg). On the other hand, in the regulation evaluation period (150 s to 300 s), the IAE using the regulation-optimized controllers is superior to that obtained using the servo-optimized controllers.

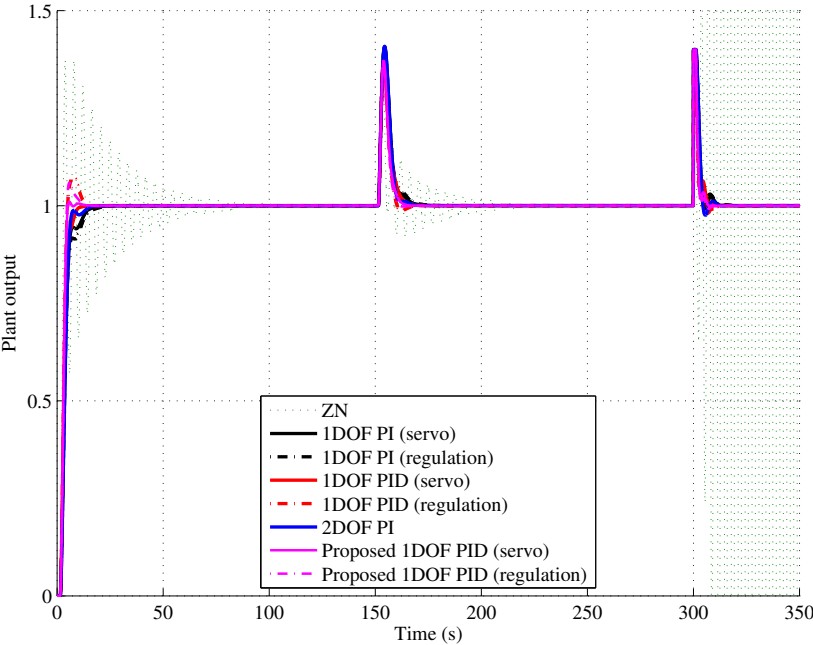

**Figure 16.** Plant outputs of the proposed and conventional methods.

**Table 10.** Integral of absolute error (IAE) performance.

| Design Method | IAE/$10^3$ (Servo) (0 s–150 s) | IAE (Regulation) (150 s–300 s) |
| --- | --- | --- |
| ZN [2] | 3.45 | 1028 |
| 1DoF PI (servo) [18] | 1.64 | 819 |
| 1DoF PI (reg) [18] | 1.71 | 855 |
| 1DoF PID (servo) [18] | 1.38 | 688 |
| 1DoF PID (reg) [18] | 1.40 | 596 |
| 2DoF PI [19] | 1.55 | 771 |
| Proposed 1DoF PID (servo) | 1.24 | 609 |
| Proposed 1DoF PID (reg) | 1.30 | 569 |

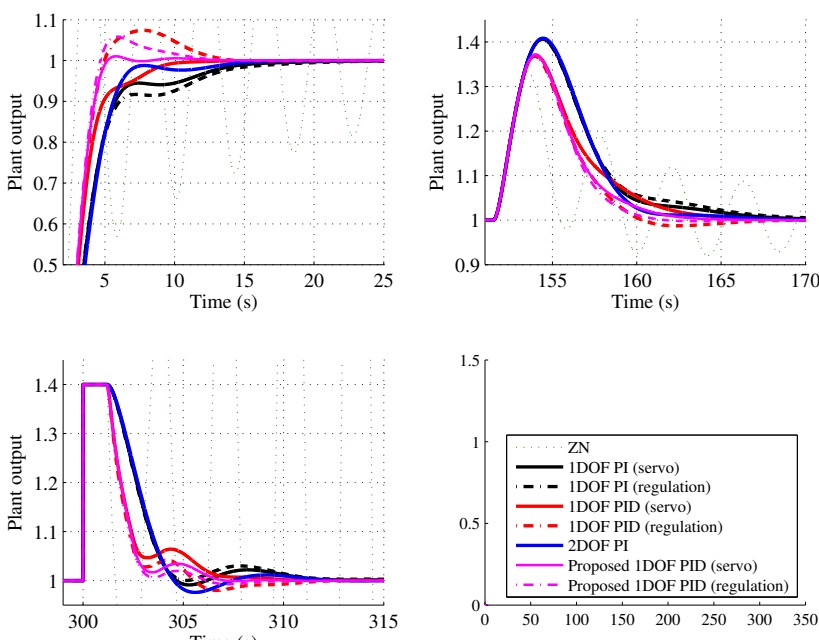

**Figure 17.** Enlarged views of Figure 16.

In the servo evaluation period, both of the proposed controllers, i.e., the proposed 1DoF PID (servo) and the proposed 1DoF PID (reg), are superior to all of the conventional controllers. On the other hand, in the regulation evaluation period, the proposed regulation-optimized controller, i.e., 1DoF PID (reg), is superior to all of the conventional methods, and the proposed servo-optimized controller, i.e., 1DoF PID (servo), is better than the conventional controllers, except for the conventional regulation-optimized PID controller, i.e., 1DoF PID (reg). The reason the proposed controllers are superior to the conventional SOPDT-based controllers is that the conventional methods are designed based on quantized values for *a*, whereas in the proposed method, the quantized values are interpolated, and hence the approximation error is improved.

## 5. Conclusions

The present study has discussed a unified trade-off design method for controlling the SOPDT system. In the conventional design method [18], the ratio of two time constants was quantized. For the purpose of interpolating the quantized ratio, the present study has extended the conventional method, and an interpolated optimal trade-off design method is proposed. The proposed method is compared with conventional methods through numerical examples, and the effectiveness of the proposed method is confirmed.

**Author Contributions:** Conceptualization, T.S., I.H., and Y.H.; methodology, T.S., Y.H., and R.V.; software, I.H. and Y.H.; validation, T.S. and R.V.; formal analysis, I.H. and Y.H.; investigation, T.S. and R.V.; resources, I.H. and Y.H.; data curation, T.S., I.H., and Y.H.; writing—original draft preparation, T.S.; writing—review and editing, R.V.; visualization, T.S., Y.H., and R.V.; supervision, T.S. and R.V.; project administration, T.S., R.V., and Y.K.; funding acquisition, T.S., R.V., and Y.K.

**Funding:** This work was supported by the the Spanish MINECO/FEDER grant DPI2016-77271.

**Acknowledgments:** The present study was supported by JSPS KAKENHI Grant Number 16K06425.

**Conflicts of Interest:** The authors declare no conflict of interest.

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
