# Peer review of "Optimal Robust PID Control for First- and Second-Order Plus Dead-Time Processes"

_applsci, doi:10.3390/app9091934_

Round 1
Reviewer 1 Report
In this paper, the authors described a robust PID control method for a second-order plus dead-time process for both servo and regulation control tasks simultaneously. The paper is written well and the references are cited properly.
Minors:
(1) Term Ms in Eq. (5) is a complex number, an abs operation abs is missing?
(2) The detail procedure of the proposed constrained optimization approach should be described.
(3) What is the prescribed reference input r(t) used during the servo optimization process?
(4) What is the disturbance d applied during the regulation optimization process?
(5) Besides the proposed look-up table results, does it exist an on-line optimization method for both servo and regulation optimization?
Author Response
Minors:
(1) Term Ms in Eq. (5) is a complex number, an abs operation abs is missing?
(2) The detail procedure of the proposed constrained optimization approach should be described.
(3) What is the prescribed reference input r(t) used during the servo optimization process?
(4) What is the disturbance d applied during the regulation optimization process?
(5) Besides the proposed look-up table results, does it exist an on-line optimization method for both servo and regulation optimization?
Answer:
(1) Thank you for your comment. That was my mistake, and an absolute operation is needed. The right hand side of Eq.(8) is corrected.
(2) More detailed procedure is added before subsection 3.1.
(3) and (4) The information is added before subsection 3.1.
(5) Thank you for your comment. There are some optimization approaches indeed. However, the constraint optimization problem must be solved whenever the optimal controller parameters are demanded, and that are not suitable for for industrial engineers. On the other hand, optimal controller parameters are easily obtained using our proposed look-up table approach. Therefore, the proposed approach is needed.

Reviewer 2 Report
The paper presents an improved method for PID controller parameter tuning. The paper is well written and the derivations are clear.
My only suggestion is that it would be useful to compare the performance against the widely known Ziegler-Nichols tuning method, in addition to those in the authors’ previous work, and to briefly discuss the relative merits of the new approach.
Author Response
Comments:
My only suggestion is that it would be useful to compare the performance against the widely known Ziegler-Nichols tuning method, in addition to those in the authors’ previous work, and to briefly discuss the relative merits of the new approach.
Answer:
Thank you for your good comment. In the revised paper, the control result using ZN method is added, and it is compared with the proposed method.
